# Detection Methods for Aflatoxin M1 in Dairy Products

**DOI:** 10.3390/microorganisms8020246

**Published:** 2020-02-12

**Authors:** Andreia Vaz, Ana C. Cabral Silva, Paula Rodrigues, Armando Venâncio

**Affiliations:** 1CEB—Centre of Biological Engineering, University of Minho, 4710-057 Braga, Portugal; andreia.sgvaz@gmail.com (A.V.); anacarolmcs@hotmail.com (A.C.C.S.); 2CIMO—Mountain Research Center, Bragança Polytechnic Institute, Campus de Santa Apolónia, 5300-253 Bragança, Portugal; prodrigues@ipb.pt

**Keywords:** Aflatoxin M1, quantification, milk, dairy products, analytical techniques

## Abstract

Mycotoxins are toxic compounds produced mainly by fungi of the genera *Aspergillus*, *Fusarium* and *Penicillium*. In the food chain, the original mycotoxin may be transformed in other toxic compounds, reaching the consumer. A good example is the occurrence of aflatoxin M1 (AFM1) in dairy products, which is due to the presence of aflatoxin B1 (AFB1) in the animal feed. Thus, milk-based foods, such as cheese and yogurts, may be contaminated with this toxin, which, although less toxic than AFB1, also exhibits hepatotoxic and carcinogenic effects and is relatively stable during pasteurization, storage and processing. For this reason, the establishment of allowed maximum limits in dairy products and the development of methodologies for its detection and quantification are of extreme importance. There are several methods for the detection of AFM1 in dairy products. Usually, the analytical procedures go through the following stages: sampling, extraction, clean-up, determination and quantification. For the extraction stage, the use of organic solvents (as acetonitrile and methanol) is still the most common, but recent advances include the use of the Quick, Easy, Cheap, Effective, Rugged, and Safe method (QuEChERS) and proteolytic enzymes, which have been demonstrated to be good alternatives. For the clean-up stage, the high selectivity of immunoaffinity columns is still a good option, but alternative and cheaper techniques are becoming more competitive. Regarding quantification of the toxin, screening strategies include the use of the enzyme-linked immunosorbent assay (ELISA) to select presumptive positive samples from a wider range of samples, and more reliable methods—high performance liquid chromatography with fluorescence detection or mass spectroscopy—for the separation, identification and quantification of the toxin.

## 1. Introduction

Mycotoxins are secondary metabolites produced naturally by moulds, and which may contaminate food and feed. Among the chemical contaminants, mycotoxins play a major role [1,2,3,4]. According to Eskola et al. [4], more than 25% of the world’s agricultural production is contaminated with mycotoxins above the EU and Codex limits.

Aflatoxins (AFs) are considered the most toxic metabolite produced by fungi, such as *Aspergillus flavus*, *Aspergillus parasiticus* and the rare *Aspergillus nomius*. The major AFs are aflatoxin B1 (AFB1), aflatoxin B2 (AFB2), aflatoxin G1 (AFG1) and aflatoxin G2 (AFG2). Aflatoxins are produced in different grains and nuts, e.g., cocoa, copra, corn, cottonseed, dried fruits, fruits, oilseeds, peanuts, pistachio nuts, sorghum, and spices in the field and during storage [5,6].

Aflatoxins have carcinogenic, hepatotoxic, teratogenic and mutagenic effects in humans and animals, even at very small concentrations, and exposure may occur through ingestion, inhalation or absorption through the skin. AFB1 is known to be the most potent, naturally-occurring carcinogen, and has been linked to liver cancer and several other maladies in animals and humans [1,3,5], being listed as a group I carcinogen by the International Agency for Research on Cancer [6,7].

Approximately 0.3% to 6.2% of the ingested AFB1 is converted to the monohydroxy derivative aflatoxin M1 (AFM1—Figure 1) in the liver of lactating animals, by the action of cytochrome P 450, and is secreted in milk [3,6,8]. Due to the binding of AFM1 to the milk proteins, particularly with casein, it can be present in dairy products, such as cheese and yoghurt [5,9]. 

Even though it is less toxic than AFB1, AFM1 has hepatotoxic and carcinogenic effects, and is relatively stable during milk pasteurization, storage, and processing [5]. Therefore, AFM1 presence in milk and in dairy products is a major risk to humans because these products are regularly consumed in the daily diet [3,6].

Aflatoxins are regulated in more than 80 countries, but their legislation is not harmonized at the international level. Therefore, the allowed limit for AFM1 in milk and in dairy products ranges from 0 to 1.0 µg/kg (Table 1) [5,6]. In the European Union (EU), AFM1 in raw milk, heat-treated milk and milk for the manufacture of dairy products should not exceed 0.05 µg/kg for adult consumption, and 0.025 µg/kg for food products meant for infants and young children [10]. In the other hand, the Codex Alimentarius Commission, and USA (United States) defined the limit of 0.5 µg/kg for milk. The EU has the most stringent regulations for mycotoxins in food [3,6,8,11].

Due to the toxicity of this molecule and considering the current maximum residue levels set, its detection and quantification is extremely important. The accurate and rapid analysis of mycotoxins has been a topic of interest for many researchers. Different analytical methods have been developed, having different levels of sensitivity and accuracy which could be used for different purposes [2,5].

The objective of this work is to provide a general overview of the different methodologies to detect and quantify AFM1 in milk and in dairy products.

## 2. Analytical Methods

The purpose of the aflatoxin analysis restricts the method choice, being important to define the level of detail of the analysis: -if it is intended to detect the presence of the analyte in the sample, rapid methods can be used;-if it is intended to detect and quantify the analyte, quantitative methods must be used.

Usually, the analytical procedures follow the following steps: sampling, sample preparation (which can include extraction and clean-up (purification)), and analysis (identification and quantification) [5,12,13].

### 2.1. Sampling

Sampling and sample preparation are of utmost importance to determine aflatoxins at the parts-per-billion level. The sampling step specifies how the sample will be collected from the bulk lot as well as its size [13]. It is important to obtain a representative sample, which involves the collection of different subsamples to obtain an aggregate sample which is as representative as possible of the entire lot. The variability associated with a mycotoxin analyses reduces by increasing the sample size, degree grinding, subsample size, and the number of aliquots quantified [13].

The European Commission Regulation 401/2006 [16] lays down the methods of sampling and analysis for the official control of the levels of AFM1 in milk. The sampling determines that an aggregate sample of at least 1 litre (or kg) should be prepared, from the collection of three to 10 incremental samples of at least 100 g.

In the case of milk, due to its liquid nature, there is less uncertainty in aflatoxin measurement. In the case of solid commodities, as cheese, mycotoxins are usually not evenly distributed, as they can accumulate in ‘hot-spots’. It is thus important to obtain a homogeneous sample. So, the entire aggregate sample must be ground and thoroughly mixed for the analytical test portion to have the same concentration of toxin as the original sample. 

### 2.2. Quantitative Methods

#### 2.2.1. Sample Preparation

In dairy products, AFM1 extraction is challenging and, typically, requires pre-treatment steps in order to remove fat and other impurities. So, centrifugation and filtration represent key stages in the extraction process [17]. For milk samples, these two processes are usually sufficient before cleaning and determination, and no use of solvents is required [18,19]. For cheese samples, preparation may also involve a slurry preparation [20]. After proper sampling and sample pre-treatment, different steps of extraction and clean-up are essential to determine and quantify the analyte of interest. Extraction and clean-up may be performed in one step [21] or in separate steps [19]. 

#### 2.2.2. Extraction

The purpose of extraction is to remove mycotoxin from the matrix, in a suitable solvent (chloroform, dichloromethane) or mixture of solvents (aqueous mixtures of polar organic solvents as acetone, acetonitrile, methanol), for partial purification and later determination. Aqueous mixtures are being increasingly used, not only for environmental reasons, but also because of their higher compatibility with subsequent clean-up steps [3,5]. 

The AFM1 extraction from cheese is generally carried out using organic solvents, like an acetonitrile–water mixture [22] or a methanol-water mixture [23]. However, chlorinated organic solvents like dichloromethane [17] and chloroform [24] can also be used. The use of organic solvents usually requires a subsequent clean-up step before determination.

Innovative methods were developed by Pietri et al. [9], based on an enzyme-assisted extraction using a cocktail of different enzymes (pepsin or pepsin–pancreatin). By using this approach, a liquefied cheese sample was obtained. In the work of Pietri et al. [9], this extract was further analysed as a milk sample (applied in Immunoaffinity Columns (IACs), and determined by HPLC–FLD). 

In comparison with the classical extraction with chloroform, the extraction of enzymes was simpler, avoiding partition in a separating funnel, solvent evaporation, and dissolution; it resulted in higher recoveries, comparable LOD (limit of detection) and LOQ (limit of quantification), and more accurate results. In addition, the method did not use chlorinated solvents, resulting in a considerable environmental advantage.

Extraction with solid sorbents (such as solid phase extraction (SPE), magnetic-solid phase extraction (MSPE)), or the Quick, Easy, Cheap, Effective, Rugged, and Safe method (QuEChERS) are alternatives to the use of organic solvents. 

In SPE, the sample is loaded on a cartridge packed with a suitable adsorbent material, on which the adsorption of the analytes to be detected takes place, and then separated by elution with a solvent, usually an organic solvent. This technique provides the extraction and purification of the analyte of interest before instrumental analysis [3,5,12]. Application of SPE column was used by different authors to extract AFM1 from milk and cheese samples, as shown in Appendix A. 

A simple and sensitive method using MSPE followed by spectrofluorimetric detection was developed by Hashemi et al. [25] for separation and determination of AFM1 in milk. This method was based on the extraction of AFM1 using magnetic nanoparticles (MMNPs), coated with 3-trimethoxysilyl-1-propanethiol (TMSPT) and modified with 2-amino-5-mercapto-1,3,4-thiadiazole (AMT). 

Alternatively to SPE, Molecularly Imprinted Polymers (MIP) are synthetic materials with recognition sites which specifically bind target molecules in mixtures with other compounds [26]. In a MIP-based SPE, the imprinted bulk polymer is packed in a cartridge, column, or extraction well plates (for high throughput analysis). This process is fast, consumes less solvent and enables the selective clean-up of the analytes [27]. The MIPs, in contrast to classical SPE sorbents, are more selective and allow the elution of analytes from the cartridges, virtually free from co-extracted compounds (Figure 2) [26]. 

Díaz-Bao et al. [28] developed a quick and easy method for the fabrication of magnetic molecularly imprinted stir-bars (MMIP-SB). They used a combination of imprinting technology and magnetite, for the analysis of AFM1 in milk powder (infant formulas). The method had a recovery of 60%, and a limit of quantification of 0.001 µg/kg. However, this methodology is still recent and more studies about its applicability in dairy products are required.

QuEChERS can be used as an alternative to other extraction methods. It was originally developed for pesticide residues from fruits and vegetables by Anastassiades, Lehotay, Stajnbaher, and Schenck [29]. However, due to its simplicity, it has been adapted for other analyses [21,30,31].

The QuEChERS method includes two steps—a simultaneous extraction and partitioning step using acetonitrile and salts, followed by a clean-up step based on a dispersive solid-phase extraction (dSPE) [30,32]. Following the acetonitrile extraction step, different sorbents, such as octadecyl silica (C18), primary secondary amine (PSA), and graphitized carbon black (GCB), can be used for additional purification, which can result in satisfactory recoveries due to the reduced matrix effect [31]. C18 is used to remove long chain fatty compounds, sterols and proteins; PSA is efficient in the removal of sugars, fatty acids and organic acids; and GCB is a strong sorbent for removing pigments, as chlorophyll, polyphenols and other polar compounds [33]. Therefore, sorbents for clean-up must be selected in accordance with the composition of the sample to be extracted. In some cases, more than one type of sorbent, such as PSA and C18, may be applied for sample cleaning, as with milk samples [21,34].

Michlig et al. [21] and Rodríguez-Carrasco et al. [34] used the QuEChERS method as a preliminary step for AFM1 determination in raw milk and its performance was in compliance with current MRLs of most dairy products regulations and with EU validation guidelines. Thus, this method could replace or complement IAC approaches, enhancing throughput and decreasing costs, for an improved monitoring of AFM1.

#### 2.2.3. Clean-Up

The main objectives of the clean-up step are the elimination of the matrix interferences and analyte preconcentration [3]. Usually, clean-up is applied after extraction to get more accurate and precise results, but it is not always required [22,35]. For most of the rapid methods based on immunochemical techniques, the diluted extracts can be used directly for analysis [36,37].

Currently, commonly used purification methods employ IAC or one-step multifunctional clean-up columns (Mycosep^TM^). These techniques provide several advantages. One example is the analysis of milk samples, after pre-treatment to defat but without any extraction step, directly in the IAC columns for analysis of the AFM1 content. However, in the case of viscous or solid samples (cheese) it is always necessary to have an extraction step.

Immunoaffinity columns are a very efficient technique of purification: they are based on the recognition of the toxin by a specific antibody. Although IACs are easy to use and have high selectivity, they are single use because of the denaturation of antibodies during the elution step, and as such the costs are high. SPE is cheaper than IAC [3,5,12].

The Mycosep^TM^ multifunctional clean-up columns consist of a number of adsorbents (charcoal, celite, ion exchange resins and others) packed in a plastic tube. Most interfering substances are retained by those adsorbents, whereas the analyte elutes without significant affinity to the packing material. Mycocsep^TM^ columns efficiently remove matrix components and can produce a purified extract within a short time, sample purification is achieved in 10 to 30 s. This rapid and efficient purification supersedes and represents an alternative to the conventional SPE or IAC methods, which typically require three to four steps [6,12].

The big difference between the Mycosep^TM^ columns and the SPE columns or IACs is that in Mycosep^TM^ columns, the analyte is eluted and the interfering compounds are retained, while in the other columns the analyte is retained (Figure 3) [6,12]. However, the IAC method has proved to be a robust technique for the separation, purification and concentration of AFM1 in dairy products [3,23,38,39].

Iha et al. [23] developed and validated a method using IAC as a clean-up column for the determination of AFM1 in cheese, yogurt, and dairy beverages. The recoveries of AFM1 ranged from 61% to 86% without correction for water content and between 67% and 101% with correction for water content. In addition, the RSDr was in the range of 2% to 12%. Thus, the performance of this method was found to be similar to that of the Association of Official Analytical Chemists (AOAC) Official Method for AFM1 in milk [40].

#### 2.2.4. Quantification

After the extraction and clean-up steps (when applied), aflatoxin must be quantified. During the past decade, several methods have been used or developed for quantification of AFM1 in dairy products [41].

Thin-layer chromatography (TLC) is a standard AOAC method for aflatoxin analysis since 1990. It is widely used in laboratories throughout the world for the qualitative analysis and quality control of food products [2,6,11].

High-performance thin-layer chromatography (HPTLC) evolved from TLC. The most important differences between TLC and HPTLC are the different particular sizes of the stationary phase, the care used to apply the samples, and the way to process the obtained data [3,11]. Another variation in TLC is the use of overpressured-layer chromatography (OPLC), which is a technique designed to integrate the benefits of HPLC and TLC [42].

The weakness of the TLC method is that it is challenging in the determination of mycotoxin concentrations with exactness [41]. So, despite screening methods based on TLC being applied on a large scale for AFM1 in milk [43,44], these are used in only a few laboratories because they do not provide an adequate LOQ [3,11].

Filazi et al. [45] analysed fifty samples of cheese for the occurrence of AFM1 using TLC, as a semi-quantitative method. The presence of AFM1 was detected in concentrations between 0.02 and 2 µg/kg in 14 of 50 samples (28%); the lowest detection limit of the method was 0.02 μg/kg and the recovery was 85.6%. The amount of aflatoxin was estimated visually by comparing with standards, and the identity was confirmed by derivatization with trifluoroacetic acid. Altogether, five cheese samples (10%) were found to have levels that exceeded the legal limits of 0.250 µg/kg established by the Turkish Food Codex.

In official control, methods of analysis for determining AFM1 in milk should be able to detect traces of AFM1 at the level of ng/kg. This performance criterion has been accomplished using the IAC column purification step, followed by LC separation and fluorescence detection. Eventually, after extensive interlaboratory testing, the method was standardized in the standard ISO 14501:2007 [3].

Liquid chromatography (LC) is a method that provides good sensitivity, high dynamic range and versatility. Detection by LC is usually made by fluorescent detection (FLD), UV absorption or mass spectrometry detection [11].

Liquid chromatographic methods for aflatoxins determination include both normal and reverse-phase separations. However, most current methods use reverse-phase HPLC, with mixtures of methanol, water and acetonitrile as mobile phases [11,46]. Reverse-phase eluents quench the fluorescence of mycotoxins like AFM1; for this reason, chemical derivatization can be necessary, using pre- or post-column derivatization. The pre-column derivatization uses trifluoroacetic acid (TFA) to forms the corresponding hemiacetals (Figure 4), while the post-column derivatization makes use of a reactive halogen like iodine or bromine [3,11,25].

A post-column derivatization method analytically equivalent to iodination and bromination is the photochemical one, being based on the formation of hemiacetals of AFM1 under UV light (Figure 4) [11,18].

Manetta et al. [35] used an HPLC method with fluorescence detection using post-column derivatization with pyridinium hydrobromide perbromide to determine AFM1 in milk and cheese. The detection limits for milk and cheese were 0.001 µg/kg and 0.005 µg/kg, respectively; and the average recoveries were 90% and 76%, respectively. Also, the precision (RSDr) ranged from 1.7% to 2.6% for milk and from 3.5% to 6.5% for cheese. The method tested proved to be simple and easily automatable, and therefore useful for the accurate and precise analysis of AFM1 in milk and cheese.

Shuib et al. [18] described the determination of AFM1 in milk and dairy products using IAC and HPLC with photochemical post-column derivatization and fluorescence detection. They reported a reduction in LOD and LOQ of about one third with derivatization, achieving 0.0085 μg/L and 0.025 μg/L, respectively. These limits were further improved when the IAC eluate was evaporated and reconstituted with mobile phase (to 0.002 and 0.004 μg/L, respectively). The method was statistically validated, showing a linear response (R^2^ > 0.999), good recoveries (85.2–107.0%), and relative standard deviations (RSD) of ≤ 7%. The proposed method was applied to various types of milk and dairy products. Only two samples (10% incidence) were positive for AFM1, even though at lower levels than the Malaysian and European legislation limits.

Iha et al. [47] studied the incidence of AFM1 in dairy products from Brazil. A total of 123 samples of three different groups of dairy products, including 58 cheese samples, 53 yoghurt samples and 12 dairy drinks were purchased from grocery stores in the Ribeirão Preto-SP area. Cheese samples were classified according to their moisture and fat contents, and were analysed by aqueous methanol extraction, IAC purification, and reverse phase LC with fluorescence detection. AFM1 was detected (> 3 ng/kg (LOD)) in 49 cheese samples. Thirty-nine (39) of the cheese samples were contaminated with AFM1 in the range of 0.010 to 0.304 µg/kg. In yogurt and dairy drinks, AFM1 was detected in 47 yoghurt samples and in 10 dairy drinks, at levels ranging from 0.010 to 0.529 µg/kg, and 0.01 to 0.05 µg/kg, respectively.

The introduction of mass spectrometry (MS) and its subsequent coupling of LC have resulted in the development of many LC-MS or LC-MS/MS methods for AFM1 analysis in dairy products [22,48,49,50]. The MS technique is used for confirmation purposes because it has the advantage of producing spectra with characteristic fragmentation patterns [3]. There are several types of instruments which can be used: single quadrupole MS, triple quadrupole (MS/MS), and linear ion trap (MS^n^). Ion trap instruments are better for identification than triple quadruple instruments (higher MS^n^ power), whereas triple quadruple instruments are better for quantification, with faster scanning and higher sensitivity [5,12].

Many LC-MS or LC-MS/MS methods comprise a single liquid extraction and direct instrumental determination without a clean-up step [22]. This is possible due to the ability of the mass analyser to filter out by mass any co-eluting impurities. However, ionization suppression can occur by matrix effects, and many authors assert that LC-MS analysis would benefit from a sample preparation (clean-up) step [5,51,52].

In a recent study by Hung et al. [53], a sensitive and rapid method was developed for the simultaneous determination of AFM1, ochratoxin A (OTA), zearalenone and *α*-zearalenone in milk by ultra-high performance liquid chromatography combined with electrospray ionization triple quadrupole tandem mass spectrometry (UHPLC–ESI–MS/MS). The LOQ for the mycotoxins were in the range of 0.003–0.015 µg/kg. The high correlation coefficients (*R*^2^ ≥ 0.996) obtained in the range of 0.01–1.00 µg/kg of the mycotoxins, along with the good recovery (87.0–109%), repeatability (3.4–9.9%) and intra-laboratory reproducibility (4.0–9.9%) at the concentrations of 0.025, 0.10 and 0.50 µg/kg, suggest that the method is adequate for simultaneously determining AFM1, OTA, zearalenone and α-zearalenone in milk [3].

The level of AFM1 was investigated in 54 samples of white and hard types of cheese produced in Serbia using ultra-high performance liquid chromatography coupled with tandem mass spectrometry (UHPLC-MS/MS). Sample extraction was performed by different methodologies for for white and for hard cheese samples. White cheese samples were prepared by crude extraction, i.e., the sample extraction was done with acetonitrile/water mixture (86:14, *v*/*v*) and the extracts were passed through a syringe filter before injection without a sample cleaning step. On the other hand, the extraction for hard cheese samples was performed with dichloromethane and acetone, followed by a SPE step prior to analysis by UHPLC-MS/MS. The average recoveries of AFM1 were 73–111% and the precision (RSD) ranged from 7% to 9%, for the first method, whilst for the second method the average recoveries were 71–80%, with an RSD ranging from 4% to 10%. Due to different matrix effects, the LOQ were 0.125 µg/kg and 0.020 µg/kg for white and hard type of cheeses, respectively. Seven samples (13%) exceeded the maximum acceptable level of 0.25 µg/kg that has been established for AFM1 in some European countries, as shown in Table 1 [22].

### 2.3. Rapid Methods

Bioassays have become increasingly useful as a rapid screening procedure before chemical analysis for mycotoxin detection [54]. For the detection of AFM1, immunochemical screening assays are mostly used, including enzyme-linked immunosorbent assays (ELISAs), immunochemical assays involving detection by electrochemiluminicesce (ECL-IA), ELISA using fluorimetric detection, and, more recently, biosensor assays [55,56].

#### 2.3.1. Sample Preparation

Immunoassays are an analytical method based on an antibody–antigen (Ab–Ag) reaction, requiring a sample preparation step. The treatment used depends on the matrix and its complexity. In the case of milk, it is only necessary to centrifuge the samples to degrease, and the skimmed milk is used directly in the test [57]. For cheese, it is necessary additionally to use an extraction step with organic solvents, such as 70% methanol and hexane, to remove interferences in the matrix such as fat [58].

#### 2.3.2. Determination

Immunoassays use specific antibodies to detect immunogens, which contain the targeted chemical structures [55].

Among the screening methods, the ELISA has been the most used for AFM1 in different food matrices, such as pasteurized and ultra-high temperature (UHT) milk, infant formula, powdered milk, yoghurt, ice cream and cheese, due to its simplicity, sensitivity and adaptability (Appendix A) [3,12,36,59,60]. A number of commercially available ELISA kits based on a competitive immunoassay format are widely used.

There are two types of competitive ELISA: direct competitive ELISA and indirect competitive ELISA (Figure 5). In the direct competitive assays, the wells of the microtiter plate are coated with a specific antibody for the analyte under analysis. After the addition of the sample, the analyte competes with an enzyme-labelled analyte to bind with a restricted number of antibodies. After incubation, unbound compounds are washed off and a chromogenic substrate is added for colour development. The measurement is made photometrically in an ELISA reader. The enzymatic activity in each well is inversely proportional to the aflatoxin concentration in the sample, i.e., the lower the absorbance, the higher the aflatoxin concentration. This happens because the higher the concentration of mycotoxin, the less the conjugate (enzyme-labelled analyte) will react with the bound antibody, leading to fainter colour development [12,35,42]. In the case of the indirect competitive assay, the analyte or its analogue, conjugated with a macromolecular carrier (e.g., BSA - Bovine Serum Albumin or OVA - Ovalbumin) is coated onto the well in the microtiter plate during incubation. When the sample and the specific antibody are added to each well, the immobilized analyte and the analyte present in the sample will compete for the antibody in solution. After a washing step, the amount of bound specific antibody is detected, through a secondary antibody, labelled with an enzyme. This approach makes it possible to simplify immunoreagents preparation because there are commercially available enzyme-labelled secondary antibodies (e.g., labelled with horseradish peroxidase (HRP) or alkaline phosphatase (AP)). However, it includes an additional step that can be eliminated by direct labelling of the specific antibody [42]. In aflatoxin analysis, direct competitive ELISAs are usually used [12]. Tavakoli et al. [61] used this method to determine the occurrence of AFM1 in 50 white cheese samples. Aflatoxin M1 was found in 60% (30/50) of the cheese samples, ranging from 0.0409 to 0.374 µg/kg.

Despite its simplicity, ELISA shows some disadvantages, such as long incubation periods and several washing and mixing stages. Based on this, in recent years several modified ELISA methods have been developed for the improved detection of AFM1 in milk and dairy products [3,55].

Vdovenko et al. [62] developed a competitive immunological assay of chemiluminescence (CL ELISA) for detection of AFM1. To improve the method’s sensitivity, a mixture of 3-(10-phenotiazine)-propane-1-sulphonate (SPTZ) and 4-morpholinopyridine (MORPH) was used to increase peroxidase induction. The limit of detection and the dynamic working range were 1.0 × 10^−6^ and 2.0 × 10^−6^–7.5 × 10^−6^ µg/mL, respectively; so, a 20-fold dilution of the milk samples was required. This prevented interferences from the milk matrices and allowed the measurement of AFM1 at concentrations that were below the maximum limit accepted. The recovery range was between 81.5% and 117.6% for the within assay and 86% and 110.6% for the between assay [3,55]. Considering that the recovery is higher than the recommended maximum limit of 110% [16], a method of optimization is still required.

Chemiluminescent detection allows the use of 384-well plates with an assay volume of 20 μL, when compared with the conventional 96-well microtiter format. This method has advantages over the conventional 96-well microtiter format, due to a 5-fold reduction in antibody, labelled probe and chemiluminescent mixture volume, which allows one to reduce the costs of the assay, maintaining the analytical performance [5].

Similarly, to increase its sensitivity, efficiency and easiness of manipulation, Kanungo and Bhand [63] developed an ELISA using fluorimetric detection. This was performed in a 384-well microplate, in which there were AFM1-specific monoclonal antibodies and secondary conjugated antibodies. AFM1 was detected at a level of 0.001 µg/L in a testing volume of 40 µL [3].

Generally, the term biosensors refers to a small, portable and analytical device based on the combination of recognition biomolecules, like antibodies and nucleic acids, with an appropriate transducer, that is able to selectively detect chemical or biological materials with a high sensitivity (Figure 6) [11].

The detection principle is the binding of the analyte of interest to the complementary bio-recognition element immobilized on a suitable support medium. After binding, a specific interaction occurs, which results in a change in one or more physico-chemical properties (e.g., pH, electron transfer, mass, or heat), which are detected with the aid of a transducer. Depending on signal transduction, biosensors can be divided into three different groups: electrochemical transducers, which rely on an electrical signal measurement (amperometric, potentiometric, and conductometric) generated by a physico-chemical change; optical transducers, in which an optical signal (colour or fluorescence) changes as a result of formation of a complex; and piezoelectric transducers, which detect changes in mass. The most common transducers in AF detection are the electrochemical and optical ones.

The immunosensors, a type of biosensor, consist of a pair of electrodes (measuring and reference), implemented using the screen-printing technique. The measurement electrode is coated with specific antibodies, which retain the aflatoxins of interest in the sample, while the other electrode (reference) is commonly made of a combination of Ag/AgCl. The measurement procedure is similar to that carried out by the ELISA test [1,3,11,42]. In this category of techniques, Rameil et al. [64] developed a potentiometric AFM1-immunosensor using 3-(4 hydroxyphenyl)propionic acid (p-HPPA) as the electron donating compound for horseradish peroxidase (HRP; EC 1.11.1.7). The assay system consisted of a polypyrrole-surface-working electrode coated with a polyclonal anti-M1 antibody (pAb–AFM1), an Ag/AgCl reference electrode, and an HRP–aflatoxin B1 conjugate (HRP–AFB1 conjugate). The optimized assay had a detection limit of 0.04 µg/L and allowed the detection of 0.5 µg/L (FDA limit) AFM1 in pasteurized milk and UHT-milk, containing 0.3–3.8% fat, within 10 min, without any sample preparation. The working AFM1 range was between 0.25 and 2 µg/L. In addition, the use of p-HPPA has the advantage of low toxicity and does not require the presence of organic solvents in the substrate buffer.

Other studies on the application of this methodology for the quantification of AFM1 in milk have been performed (Appendix A) [65,66].

The aptasensor is a particular class of biosensor and can be a good alternative to immunosensors because it is easier to synthesize and modify with a variety of chemical groups [67,68]. The difference of this sensor is in the biological recognition element, which is an aptamer instead of an antibody. The aptamer consists of a synthetic oligonucleotide ligand (either single stranded DNA (ssDNA) or RNA), generally comprising less than 80 nucleotides with a size lower than 25 kDa, and is known to exhibit high specificity and strong binding affinity [67,69,70,71]. In an aptasensor, the aptamer recognizes the molecular target against which it was previously selected in vitro. So, the aptamer can bind to a wide range of target molecules, such as drugs, proteins, toxins or other organic or inorganic molecules, with high affinity and specificity [71,72]. Different studies on the application of this methodology for the quantification of AFM1 in milk have been performed (Appendix A).

Biosensors have the advantage of being simple, rapid, cost effective and portable devices that are specific to the target mycotoxin. However, their sensitivity and stability still need improvement to allow long-term use [2,5].

## 3. Conclusions

Due to the ingestion of feed contaminated with AFB1, lactating animals may secrete AFM1 in milk. Thus, dairy products, such as milk, cheese and yogurts, are therefore potentially contaminated with this toxin, which, although less toxic than AFB1, also exhibits hepatotoxic and carcinogenic effects, and is relatively stable during pasteurization, storage and processing. The presence of AFM1 in dairy products poses a major risk to the health of consumers, as these products are largely consumed. For this reason, the establishment of maximum limits allowed in this type of product fostered the development of methodologies that allow its detection and quantification in dairy products.

There are several methods for the detection of AFM1 in dairy products and the choice of the method should consider the following factors: sensitivity, precision and reliability of the method, to ensure compliance with current standards. Analysis of AFM1 is based on the following steps: sampling, sample preparation, extraction, clean-up, and analysis (identification and quantification).

Regarding the extraction stage, this is responsible for the maximum extraction of the mycotoxin from the food matrix. Given the most recent development at this stage, it is concluded that the method proposed by Pietri et al. [9], based on the use of proteolytic enzymes (pepsin or pepsin-pancreatin), may be a good alternative to the current methods for the analysis of cheeses, since it avoids the use of organic solvents.

After extraction and before quantification, a purification stage may be required. Given that milk and cheese are complex food matrices, and this stage is responsible for the reduction/elimination of interferences, purification allows the increase in the sensitivity of the method. Usually, immunoaffinity columns are used, and IAC-based methods have proved to be robust for the purification, separation and concentration of AFM1 in milk and in dairy products. Recent developments based on MIPs and aptamers have not yet reached a position of being competitive against IACs.

After the latter two steps, the quantification of mycotoxin is performed. There are numerous methods that allow the detection and quantification of these compounds, and the most recurrent methods were divided into two groups—namely, chromatographic methods and bioassays. Both allow the detection of AFM1; however, bioassays are techniques commonly used for screening since the immunological methods may give rise to false positives. This is because, although the antibodies are specific to their antigens, they can react with other substances. Thus, the use of immunological methods, such as the ELISA, could be used at a preliminary stage to select from a wide range of samples those that are contaminated with the toxin under study. Subsequently, other methods are used to confirm the results. Among several methods cited throughout the work, it has been found that those best suited for AFM1 detection in dairy products are chromatographic ones with fluorescence detection or those coupled to mass spectroscopy.

## Figures and Tables

**Figure 1 microorganisms-08-00246-f001:**
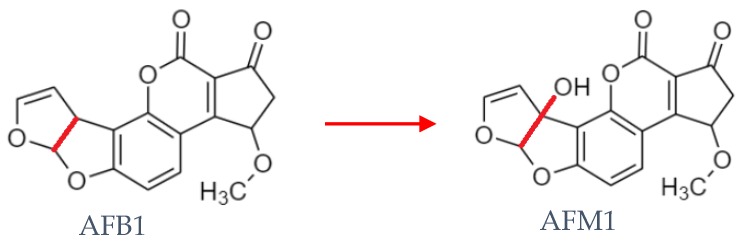
Illustration of the conversion from aflatoxin B1 (AFB1) to aflatoxin M1 (AFM1).

**Figure 2 microorganisms-08-00246-f002:**
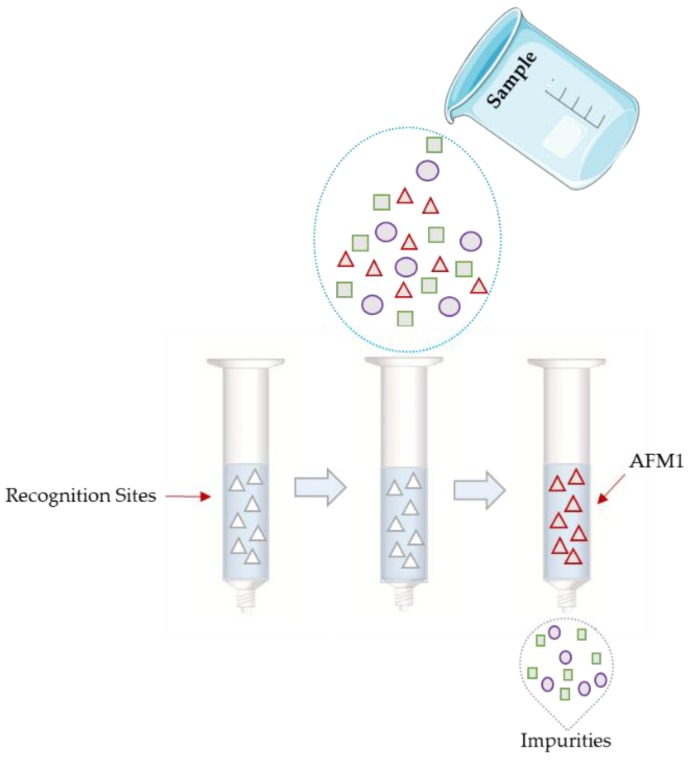
Molecular imprinting process.

**Figure 3 microorganisms-08-00246-f003:**
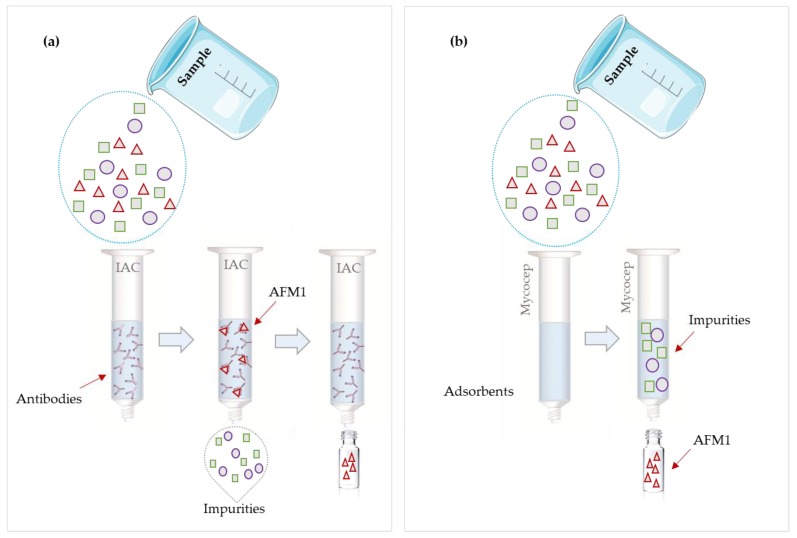
(**a**) Scheme of immunoaffinity column (**b**) Scheme of Mycosep^TM^.

**Figure 4 microorganisms-08-00246-f004:**
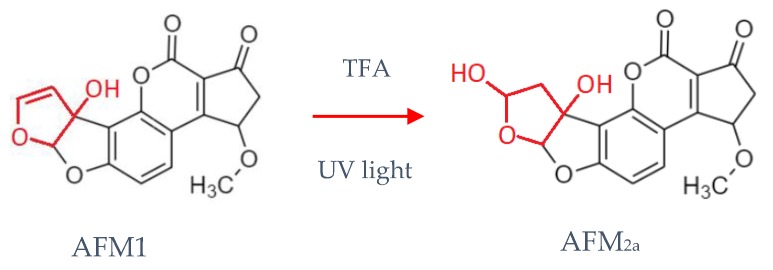
Reaction of AFM1 with trifluoroacetic acid (TFA) and UV light.

**Figure 5 microorganisms-08-00246-f005:**
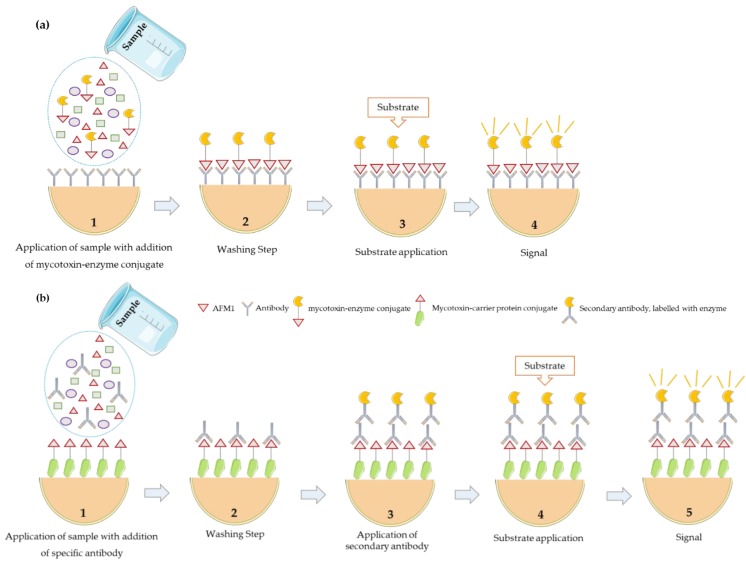
Competitive ELISA principle. (**a**) Direct format and (**b**) indirect format.

**Figure 6 microorganisms-08-00246-f006:**
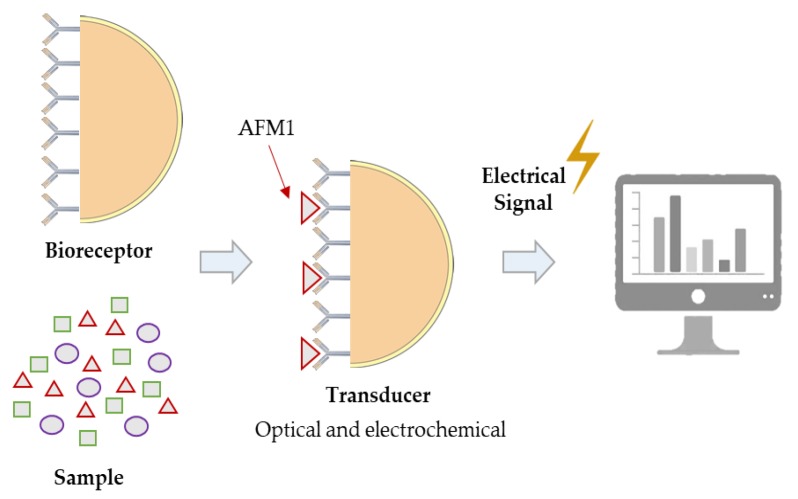
General basis parts of immunosensors.

**Table 1 microorganisms-08-00246-t001:** Regulation of aflatoxin M1 in dairy products in different countries [6,14,15].

Country	Milk (µg/L)	Dairy Products (µg/kg)
**Argentina**	0.05	0.50 (milk products); 0.25 (cheese)
**Brazil**	0.5	5 (milk powder); 2.5 (cheese)
**China**	0.5	0.5 (milk products)
**Egypt**	0	0
**EU**	0.05 0.025 (food products meant for infants and young children) **Austria** 0.01 (pasteurized infant milk) **France** 0.03 (for children ˂3 years)	**Italy** 0.25 (soft cheese); 0.45 (hard cheese) **Austria** 0.020 (butter); 0.25 (cheese); 0.40 (milk powder) **The Netherlands** 0.020 (butter); 0.020 (cheese)
**Honduras**	0.05	0.250 (cheese)
**Iran**	0.05	0.50 (milk powder); 0.020 (butter and butter milk); 0.250 (cheese)
**Nigeria**	1	-
**Switzerland**	0.05	0.25 (cheese)
**Turkey**	0.05	0.25 (cheese)
**USA**	0.5	-

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
