# Peer review of "Detection Methods for Aflatoxin M1 in Dairy Products"

_microorganisms, 2020, doi:10.3390/microorganisms8020246_

Round 1
Reviewer 1 Report
The manuscript presents well organized and comprehensive description of detection methods for aflatoxin M1 in dairy products, with wide coverage of analytical techniques, some of historical relevance, fully characterized and firmly established methods used for the official control, the newest methods still in development…
The authors neatly explained theoretical background of presented analytical techniques and provided references to relevant papers. Studies introducing innovative approaches to different stages of analytical procedure, extraction, clean-up and quantification, are identified and discussed. A separate section deals with rapid methods, presenting scientific reasoning and design of various sensors. Methods are compared in order to emphasize their advantages and disadvantages. Discussion is relevant, with justified and logical conclusions, without inconsistencies. References provide thorough and up-to-date overview of the published research covering the chosen topic.
The manuscript is well guided, easy to read, yet provides a summary of the most relevant characteristics of the methods for the detection of aflatoxin M1 in dairy products. Thus, the manuscript could be used as excellent educational material for young researchers and important professional update for experienced ones. The topic is also of interest for dairy industry, food safety authorities, all interested in public health.
Author Response
We thank the reviewers for the comments.
Tha manuscript was not changed, since this reviewer did not raise any coment of the content.
Reviewer 2 Report
The Authors presented a review paper entitled ''Detection methods for Aflatoxin M1 in dairy products''. The paper is interesting and the authors tried to address the detection methods for AFM1 in milk and milk products.
I have a few comments to be revised.
In the abstract, some missing full-length sentences for abbreviations (eg. QuECHRS, ELISA, HPLC) On figure2, please remove the words on top of the figure (Crops feed and milk), unless they will mislead the readers. on line 381, it says there are two types of ELISA. types of ELISA are not only two. please correct it ''there are two types of competitive ELISA''.Author Response
First, we thank the reviewer for the contributions toward improving this work. In the current version, we have addressed all points raised by the reviewers.
abstract was corrected figure 1 was corrected accordingly comment about line 381 was also corrected